# Fractal Geometric Model for Statistical Intermittency Phenomenon

**DOI:** 10.3390/e25050749

**Published:** 2023-05-03

**Authors:** Walid Tarraf, Diogo Queiros-Condé, Patrick Ribeiro, Rafik Absi

**Affiliations:** 1Laboratory of Research in Industrial Eco-Innovation and Energetic (LR2E), Ecole Supérieure d’Ingénieurs ECAM-EPMI, 13 Bd de l’Hautil, 95000 Cergy, France; r.absi@ecam-epmi.com; 2Laboratory of Energetics Mechanics and Electromagnetism (LEME), Université Paris Nanterre, Pôle de Ville d’Avray, 50 rue de Sèvres, 92410 Ville d’Avray, France; dqueiros-conde@parisnanterre.fr (D.Q.-C.); patrick.ribeiro@live.fr (P.R.)

**Keywords:** scale entropy, entropic skin theory, turbulence, intermittency, nonlinear dynamics, statistical physics, complex systems, multiscale fractal geometry, clustering

## Abstract

The phenomenon of intermittency has remained a theoretical concept without any attempts to approach it geometrically with the use of a simple visualization. In this paper, a particular geometric model of point clustering approaching the Cantor shape in 2D, with a symmetry scale θ being an intermittency parameter, is proposed. To verify its ability to describe intermittency, to this model, we applied the entropic skin theory concept. This allowed us to obtain a conceptual validation. We observed that the intermittency phenomenon in our model was adequately described with the multiscale dynamics proposed by the entropic skin theory, coupling the fluctuation levels that extended between two extremes: the bulk and the crest. We calculated the reversibility efficiency γ with two different methods: statistical and geometrical analyses. Both efficiency values, $\gamma_{stat}$ and $\gamma_{geo}$, showed equality with a low relative error margin, which actually validated our suggested fractal model for intermittency. In addition, we applied the extended self-similarity (E.S.S.) to the model. This highlighted the intermittency phenomenon as a deviation from the homogeneity assumed by Kolmogorov in turbulence.

## 1. Introduction

### 1.1. Statistical Intermittency: From the Theory of Kolmogorov

Kolmogorov’s theory in developed turbulence (K41) assumes that turbulence is statistically homogeneous (invariant due to spatial translation, far from limits), isotropic (invariant due to rotation), self-similar (invariant due to scale expansion) and stationary (invariant due to time translation) [1,2,3]. According to Kolmogorov’s theory, the developed turbulence can be described using the structure functions of dissipated energy $<{\varepsilon (r)}^{p}>$ and velocity $<{\delta V(r)}^{p}>$, with *ε(r)* the rate of energy dissipation per unit mass averaging over an r-dimensional ball and *δV(r)* the velocity fluctuation *V* between two points separated by a distance of *r*.

This theory, based on the hypothesis of the homogeneity of energy dissipation, predicts universal laws of scales $<{\delta V(r)}^{p}>\sim r^{z_{p}}$ and $<{\varepsilon (r)}^{p}>\sim r^{\tau_{p}}$ for high Reynolds numbers and scales in the inertial range (η $\ll r\ll r_{0}$, where η and $r_{0}$ are, respectively, the Kolmogorov scale and the integral scale), such as $z_{p}$ and $\tau_{p}$ being the scaling exponents. Considering the assumption of the refined similarity $<{\delta V(r)}^{p}>={<{\varepsilon (r)}^{p/3}>r}^{\tau_{p/3}}$, this means that $z_{p}=\frac{p}{3}+\tau_{p/3}$ [4,5,6,7]. Indeed, Kolmogorov predicted that $z_{p}$ followed a linear law, as the energy dissipation rate was assumed to be scale-independent according to the homogeneity’s hypothesis ($<{\varepsilon (r)}^{p}>={<\varepsilon (r)>}^{p}$), hence, $\tau_{p/3}=0$.

However, the Kolmogorov’s theory does not agree with the results of many numerical and experimental studies carried out later, showing significant deviations from the linearity of scaling exponents assumed by Kolmogorov [1,8,9,10,11,12,13,14]. This deviation from linearity (homogeneity’s assumption) is due to the intermittent characteristics of energy dissipation, where the gradient of turbulent flows fluctuates significantly over the volume occupied by the fluid [15,16,17]. This phenomenon is known as “intermittency”.

Intermittency in turbulence is a phenomenon of nonlinear dynamics resulting from complex and nonlinear interactions between different variables of a system [18,19,20]. This phenomenon has been studied with the use of various geometrical theories, such as fractal geometry. One of the most important was the theory of Procaccia and Grassberger on the strange attractor method [21]. Therefore, several models have been developed to describe it and determine the exponents of intermittency, such as the She and Lévêque’s model [22,23], the multifractal model [24,25], the β-model [14,26] and the log-normal model of Kolmogorov, Obukhov and others [13,24,27].

Turbulent flow has been presented as a hierarchical structure of fluctuation levels distinguished using some thresholding processes. Thus, these fluctuation levels are characterized through structure functions, such as the difference between the velocity or energy dissipation rate with average values, respectively. Therefore, the hierarchy of intermittent turbulence extends between disordered structures corresponding to low fluctuations and ordered structures, where the highest fluctuations are generated. This theoretical concept has been validated through experimental observations, such as for high Reynolds numbers, as well as for large thresholds, with the strongest fluctuations taking place on a filamentary structure of a fractal dimension equal to 1 [10,28,29], while, for a low threshold, the fluctuations take place on a structure of fractal dimension equal to 2.36. Thus, this variation in fractal dimensions with fluctuation levels emphasizes the hierarchical multiscale characteristics of intermittent turbulence [24,25]. Note that these multiscale characteristics have been discussed in several studies that proposed that turbulent interfaces, such as those in clouds, turbulent jets, boundary layers and turbulent combustion, exhibit fractal behavior for scales belonging to the range of inertial scales [30,31,32].

### 1.2. Clustering and Fractal Analysis

Clustering is a data analysis technique widely used in the field of dynamical systems, especially turbulence, to identify spatial and temporal structures in turbulent flows [33,34,35,36]. Clustering can be used to divide a flow into fields with similar characteristics in terms of turbulent dynamics. The identified fields can correspond to vortices, laminar flows and recirculation zones [37,38,39,40]. This technique tends to be used in conjunction with a fractal analysis to study the phenomenon of developed turbulence.

The fractal analysis is a geometric analysis method that quantifies the irregularity and complexity of structures in dynamic systems and their variability at different spatial scales [41,42,43]. Since turbulence is a complex and chaotic phenomenon, characterized by rapid and random fluctuations in flow variables such as velocity and pressure, the fractal analysis has been widely used to quantify the geometric complexity of turbulent structures and their variability at different spatial scales [44,45]. The complexity can be measured well with the fractal dimension, having been measured using several methods. One of the well-known methods is box counting. Based on the concept of Hausdorff dimensions, this method consists of covering the object or structure with boxes of different sizes and counting the number of boxes needed to cover the object. In this way, the fractal dimension is equal to the ratio of the logarithm of the number of boxes to the logarithm of the size of the boxes [46]. This has been used to identify the most important spatial scales for turbulent energy dissipation, such as the Kolmogorov scale [47]. The combined use of these two techniques has led to a better understanding of turbulence mechanisms.

### 1.3. Entropic Skis Geometry (E.S.G) to Describe Intermittency

One of these models, the entropic skin geometry (E.S.G.), shown to be in agreement with other experiments [17,48], is an interpretation of this intermittency phenomenon achieved through a coupling between the multiscale fractal geometry and statistical intermittent analysis within the turbulence by assuming the existence of a certain hierarchy of correlated fractal structures, characterizing the dynamics of fluctuation levels (skins). Each fluctuation level of the order p has been associated with a fluctuating field ${Ω}_{p}$, representing a structure of dimension $\Delta_{p}$, included in the field ${Ω}_{p-dp}$ of dimension $\Delta_{p-dp}$, such as ${Ω}_{p-dp}⊃{Ω}_{p}\;with\;\Delta_{p-dp}<\Delta_{p}.$ All fluctuating fields are included in the initial field ${Ω}_{0}$. Therefore, according to this theory, turbulence is considered a superposition of structures ranging from the extremely disordered one, showing less fluctuations (at p=0) with dimension $\Delta_{0}$ equal to the embedding dimension d ($\Delta_{0}=d$), to the extremely ordered one (filamentary form), where the largest fluctuations (crest) in regard to the mean occur (when p→∞). For p=1, there is dimension $\Delta_{1}$, designating the classical fractal dimension $D_{f}$, $\Delta_{1}{=D}_{f}$ and corresponding to the structure called the “bulk emergence”. Likewise, for p→∞, there is the extremely ordered structure that we called the “ crest convergence “ with crest dimension $\Delta_{\infty }$ [17,49]. Therefore, the bulk and the crest represent the two extremities of the hierarchy of skin structures.

Furthermore, at the geometric level, the concept of scale entropy $S_{p}\left(r\right)$ for each fluctuation level p in a scale r has been introduced to quantify the spatial extension of each skin in the intermittency hierarchy. It is defined as the logarithm of the space occupation rate $w\left(r\right)$ [36,50]:$$S_{p}\left(r\right)=ln\left(w\left(r\right)\right)=(d-\Delta_{p})ln\left(r/r_{0}\right)$$
where $w\left(r\right)=\frac{v_{p}(r)}{v_{0}}$ and $v_{p}(r)$ is the volume occupied by ${Ω}_{p}$ at scale r in a total volume corresponding to the integral scale $v_{0}={r_{0}}^{d}$. Since $N_{p}\left(r\right)$ is the minimum number of coverages with thickness r covering the set ${Ω}_{p}$, the volume of the set can be determined using $v_{p}(r)=N_{p}(r)r^{d}$, with $N_{p}\left(r\right)={\left(r/r_{0}\right)}^{{-\Delta }_{p}}$ showing that the set ${Ω}_{p}$ is fractal.

In addition, the entropic skin theory proposes the notion of geometric reversibility efficiency γ, defined as the scale entropy flux ratio, characterizing its transmission through two consecutive skins, p and +dp:$$\gamma_{p}\left(r\right)=\frac{S_{p+dp}\left(r\right)-S_{p}\left(r\right)}{S_{p}\left(r\right)-S_{p-dp}\left(r\right)}=\frac{\Delta_{p+dp}-\Delta_{p}}{\Delta_{p}-\Delta_{p-dp}}$$

This assumes that the reversibility efficiencies are constant throughout the whole skin hierarchy. Thus, the relationship in the reversibility efficiency across all structures of the overall hierarchy obtained with the recurrence method can be given by:$$\gamma =\frac{\Delta_{1}-\Delta_{\infty }}{d-\Delta_{\infty }}$$
such as $\gamma_{p}=\gamma^{dp}$.

For γ=1, this means that the loss of entropy through the skins is zero and, therefore, the phenomenon is nonintermittent (homogeneous). For γ→0 the phenomenon becomes infinitely intermittent.

In the statistical approach, the value of a structure function $<{\varepsilon (r)}^{p}>$ is given by its “active part”, offering two contributions; the first one $\overset{-}{\varepsilon }{\left(r\right)}^{p}$ is the average energy dissipation rate over the actual active part of the field, and the second $f_{p}(r)$ is the volume fraction, which indicates an intermittency factor and allows for the “lacunary aspect” of energy dissipation [3,20]. Thus, we would have:(1)$$<{\varepsilon \left(r\right)}^{p}>=\overset{-}{\varepsilon }{\left(r\right)}^{p}f_{p}\left(r\right)=\overset{-}{\varepsilon }{\left(r\right)}^{p}{\left(\frac{r}{r_{0}}\right)}^{d-\Delta_{p}}$$
for p=1, $<\varepsilon \left(r\right)>={\overset{-}{\varepsilon }}^{}{\left(r/r_{0}\right)}^{d-\Delta_{1}}$. 

It should be noted that $\overset{-}{\varepsilon }(r)$ follows a power law [49]:(2)$$\overset{-}{\varepsilon }\left(r\right)\sim r^{\chi }\;withe\;\chi =\Delta_{f}-d$$

Inspired by the idea of coupling between the fractal geometry and the statistics, these quantities allow to determine γ experimentally, proposing the relative structure functions:$$\mu_{p}(r)=\frac{<{\varepsilon (r)}^{p}>}{<{\varepsilon (r)}^{p-dp}>}$$
such as $<{\varepsilon (r)}^{p}>=\overset{-}{\varepsilon }{\left(r\right)}^{p}{\left(r/r_{0}\right)}^{d-\Delta_{p}}$ and $\gamma^{dp}=\frac{\Delta_{p+dp}-\Delta_{p}}{\Delta_{p}-\Delta_{p-dp}}$, through which we obtain the relationship of the relative structure function [22]:(3)$$\mu_{p+dp}\left(r\right)={\overset{-}{\varepsilon }\left(r\right)}^{\left(1-\gamma^{dp}\right)}{\mu_{p}\left(r\right)}^{\gamma^{dp}}$$

Since $\mu_{p+dp}\left(r\right)$= ${\overset{-}{\varepsilon }(r)}^{dp}{\left(r/r_{0}\right)}^{\Delta_{p}-\Delta_{p+dp}}$, for p→∞, we could obtain $\mu_{p\to \infty }(r)=\overset{-}{\varepsilon }(r)$. The same applies to the velocity fluctuation, introducing relative structure functions:$$\nu_{p}(r)=\frac{<{V(r)}^{p+dp}>}{<{V(r)}^{p}>}$$

Similarly, for the velocity, we could still write based on the relationship $\overset{-}{\varepsilon }\left(r\right)=\frac{{\overset{-}{\delta V}\left(r\right)}^{3}}{r}$ and the refined similarity hypothesis $<{\delta V(r)}^{p}>=<{\varepsilon (r)}^{p/3}>r^{p/3}$ [3]:(4)$$\nu_{p+dp}\left(r\right)={\overset{-}{\delta V}\left(r\right)}^{\left(1-{\gamma_{\nu }}^{dp}\right)}{\nu_{p}\left(r\right)}^{{\gamma_{\nu }}^{dp}}\;where\;\gamma_{\nu }=\gamma^{\frac{1}{3}},$$
with $\nu_{0}\left(r\right)=<\delta V\left(r\right)>$ and $\underset{p\to \infty }{\lim}\nu_{p}$(r) = $\overset{-}{\delta V}(r)$.

Note that this equation was also obtained by She and Lévêque in their model [22]. Thus, γ could be obtained statistically with Equations (3) and (4).

The reversibility factor calculated with the geometric analysis (using the entropic scale) would be the same when calculated with the statistical analysis (using the structure functions).

For this purpose, in order to distinguish them, we noted them, respectively, as $\gamma_{geo}$ and $\gamma_{stat}$, whose equality can be expressed through $\gamma_{stat}=\gamma_{geo}$.

This equality has an intrinsic and fundamental consequence in the theory of E.S.G, allowing to link the statistical and geometric aspects of a spatial distribution [51,52].

Note that the expression of the exponents $z_{p}=\frac{p}{3}+\tau_{p/3}$ allows, via the Legendre transformation, to obtain the multifractal spectrum. Therefore, a link between the entropic skin intermittency theory and the concept of multifractals was established [49,53].

### 1.4. Extended Self-Similarity (ESS)

According to the extended self-similarity principle proposed in developed turbulence (for high and low Reynolds numbers), any function $<{\delta V(r)}^{p}>$ of order p (either of velocity or of the rate of energy dissipation) is a power law of any other function $<{\delta V(r)}^{p}>$ of order q, whose logarithmic plot slope is $z_{p}/z_{q}$ [54].

In the case of homogeneous and isotropic turbulent flows, according to the Kolmogorov assumption, the refined similarity relation ($z_{p}=p/3$) implies that $z_{p}/z_{q}=p/q$. However, in an intermittent case, a deviation from p/q can arise due to fluctuations produced through the energy dissipation rate.

In this paper, we propose a multiscale model to describe and represent the phenomenon of statistical intermittency. This phenomenon has never been defined in a geometrical pattern, and has remained a theoretical concept without any attempts to approach it through the use of a simple visualization. Indeed, the hierarchical structure of fluctuation levels has been assumed with several models and theories (such as the She and Leveque model [22]), but this remained within the limits of a hypothesis. Note that in the search for a geometrical explanation of intermittency, Queiros-Condé (1997) showed that diffusion-limited aggregates (DLAs) displayed analogous intermittent features to fully developed turbulence [55].

Therefore, we introduce a cluster model that generates intermittency across scales in a fractal manner, modeling this phenomenon in the visualization step in order to better understand it.

We take up tools and statistics already used to describe intermittency in developed turbulence, mentioned above, such as the entropic skin theory and the extended self-similarity hypothesis, to verify the ability of this model to describe statistical intermittency [55].

The structure of this paper is presented as follows: After Section 1, Section 2 details the proposed cluster model to describe intermittency. Section 3 introduces the measures applied to perform the required geometric and statistical analyses, as well as the parameters that apparently influence them. In Section 4, we verify the validity of the model to describe intermittency by applying the entropic skin theory. The Section 5 and Section 6 include multiscale and statistical analyses, respectively. In Section 7, we discuss the results of the data analyses and validate the ability of the proposed geometric model to describe intermittency. Finally, Section 8 synthesizes the paper and discusses the next steps in the research. In the following, we present our model.

## 2. Geometric Model for Intermittency

The objective of our work was to find a multiscale geometric model for the intermittency phenomenon. For this purpose, we proposed a simple model, comprising a clustering of points taking the form of a two-dimensional Cantor with a symmetry scale θ that represented the intensity of the clustering intermittency. The model started by distributing points randomly on a plane in a homogeneous way using MATLAB software. Then, we introduced a differentiation for the density by defining zones where the density of the points would be higher. In these areas, the points would be distributed randomly, but with a higher density than in the first step, following the 2D Cantor geometry. The differentiation process was then reproduced across scales. We, thus, associated the homogeneous distribution of points with a fractal differentiation mechanism that reproduced many features of the intermittent phenomenon. This mechanism allowed for creating a clustering structure showing high fluctuations in the point number and widely differentiated local densities. The aim was to simulate the bulk–crest dynamic supposed to describe the hierarchy of intermittency in developed turbulence.

This model was iteratively developed following the steps detailed below:
Step 1: Consider a domain of size $l_{0}\times l_{0}$ pixels, throwing in a few $n^{\left(0\right)}$ points uniformly at random: iteration 0 (Figure 1a).Step 2: Remove the areas at the four corners from the initial draw (known in normal 2D Cantor), having sides of size $l_{1}=l_{0}\theta$; θ is a symmetry scale and $l_{0}$ is the entire scale. Then, throw the same number of points $n^{\left(0\right)}$ as in the previous iteration into each of these areas (also uniformly random) (Figure 1b).Step 3: Consider each generated “sub-block” square and apply this process.Step 4: Iterate to infinity.

Note that scale $l_{i}$ represents the size of the sides of the generated squares at the ith iteration, such as $l_{i+1}=l_{i}\theta$. The model is shown in Figure 1c.

Generally, we worked in a domain with 4096 × 4096-sized pixels, except when we wanted to know the effect of the size variation $l_{0}$ on the measurements.

The scaling factor θ ($\theta =\frac{l_{i+1}}{l_{i}}$) showed a fundamental factor reflecting the degree of intermittency varying in the range [0, 0.5], such that if θ was relatively low, the distribution would display strong fluctuations, and, thus, considerable intermittency. If θ was high (close to 0.5), the distribution would become close to homogeneous.

## 3. Measurements

The interest in the 2D Cantor clustering model is that it is very simple and efficient to describe, and allows for a clear visualization of the geometry of an intermittent structure regarding the bulk and crest dynamic.

To perform the required analysis, we covered the square box with a grid of the same size, having an elementary mesh length r representing the scale. We counted the different parameters at each scale *r* (see Figure 2).

For the scale analysis, we measured the number of meshes affected by the distributed points for each scale r $N_{F}(r)$ in order to represent the shape of $N_{F}(r)$ in a logarithmic scale as a function of scale r, according to the box-counting method mentioned in Section 1.3. The fractal dimension presented the opposite of the slope of the curve.

At the statistical analysis level, we counted the number $n_{i}$ of points in each cell in order to measure the local fluctuations in point numbers. We determined the mean number of points at the given scale r as follows:$$<n_{i}(r)>=\frac{1}{M(r)}\sum_{i=1}^{N(r)}n_{i}$$
where M(r) is the number of blocks at scale r and $n_{i}\;is$ the number of points in mesh i.

We, therefore, defined the structure functions (mean fluctuation rate of point numbers) at fluctuation level *p*:(5)$${<\delta n_{i}(r)}^{p}>=\frac{1}{M(r)}\sum_{i=1}^{N(r)}{\left|n_{i}-<n_{i}(r)>\right|}^{p}={<\left|n_{i}-<n_{i}(r)>\right|}^{p}>$$

According to the principle of cascading fluctuating structures through the scales of entropic skins, we assumed that the structure functions followed a scaling law:(6)$${<{\delta n}_{i}\left(r\right)}^{p}>\sim r^{z_{p}}$$

Due to of the random jets (uniform distribution), some deviations in the number of affected meshes $N_{F}(r)$ could exist. This produced uncertainty problems. For this reason, we remedied this problem by repeating the jets of points several times in order to reduce these uncertainties and by averaging the measurements on all the jets.

Several parameters seemed to influence the geometric and statistical analyses, including the symmetry scale and the topic scale $l_{*i}$, as well as the population factor a.

We defined $l_{*i}$ as the topic scale, the mean distance between particles, at a certain iteration i:(7)$$l_{*i}=\sqrt{\frac{{l_{i}}^{2}}{{an}_{b,i}}}=\frac{l_{i}}{\sqrt{{an}_{b,i}}}$$
where $n_{b,i}$ represents the number of points per block associated with iteration i, $l_{i}$ is the size of the block, such that $l_{i+1}=l_{i}\theta$ with θ is the symmetry scale (intermittency parameter), the population factor a represents the ratio of the number of points per block and $n_{b,i}$ and $n_{b,i+1}$, are associated, respectively, to two certain consecutive iterations I and i+1, such that $a=\frac{n_{b,i+1}}{n_{b,i}}$ (see Figure 3).

## 4. Application of Entropic Skin Geometry to the Fractal Model

Similar to the dynamics of entropic skins applied in developed turbulence [51,52], we assumed that the intermittent statistic in the fractal clustering model was driven by two specific fractal objects: the bulk and the crest. As such, these two objects were associated, respectively, with the fluctuating structures ${Ω}_{1}$, with dimension $\Delta_{1}$ = $D_{f}$ (at p=1), such as $D_{f}$ as the fractal dimension of the phenomena, and ${Ω}_{\infty }$, with dimension $\Delta_{\infty }$ (p→∞).

Therefore, using analogy with the assumption that giving the mean fluctuation rate of energy dissipation would be associated with the p-level of fluctuation through $<{\varepsilon \left(r\right)}^{p}>=\overset{-}{\varepsilon }{\left(r\right)}^{p}f_{p}\left(r\right)=\overset{-}{\varepsilon }{\left(r\right)}^{p}{\left(\frac{r}{r_{0}}\right)}^{d-\Delta_{p}}$(in Equation (1)), we proposed the mean fluctuation rate of point numbers in scale r, associated with the p-level, given by:(8)$${<{\delta n}_{i}\left(r\right)}^{p}>={{<{\delta n}_{i}\left(r\right)>}^{p}}_{F}{\left(\frac{r}{r_{0}}\right)}^{d-{∆}_{p}}$$
where ${{<{\delta n}_{i}(r)>}^{p}}_{F}$ is the mean fluctuation rate of point numbers on the affected blocks, corresponding to the fluctuation level p.

Likewise:(9)$${<{\delta n}_{i}\left(r\right)}^{p+dp}>={{<{\delta n}_{i}\left(r\right)>}^{p+dp}}_{F}{\left(\frac{r}{r_{0}}\right)}^{d-{∆}_{p+dp}}$$

Dividing Equation (9) by Equation (8) gave:$$\frac{{<{\delta n}_{i}(r)}^{p+dp}>}{{<{\delta n}_{i}(r)}^{p}>}={{<{\delta n}_{i}(r)>}^{dp}}_{F}{\left(\frac{r}{r_{0}}\right)}^{{∆}_{p}-{∆}_{p+dp}}$$

Let us define the relative structure function $\alpha_{p}(r)$ as follows:$$\alpha_{p}(r)=\frac{{<{\delta n}_{i}(r)}^{p}>}{{<{\delta n}_{i}(r)}^{p-dp}>}$$

Obviously:$$\alpha_{p+1}(r)=\frac{{<{\delta n}_{i}(r)}^{p+dp}>}{{<{\delta n}_{i}(r)}^{p}>}$$

Therefore:$$\frac{\alpha_{p}(r)}{{{<{\delta n}_{i}(r)>}^{dp}}_{F}}={\left(\frac{r}{r_{0}}\right)}^{{∆}_{p-dp}-{∆}_{p}}\;and\;\frac{\alpha_{p+dp}(r)}{{{<{\delta n}_{i}(r)>}^{dp}}_{F}}={\left(\frac{r}{r_{0}}\right)}^{{∆}_{p}-{∆}_{p+dp}}$$

This gave us:$$\frac{\alpha_{p+dp}(r)}{{{<{\delta n}_{i}(r)>}^{dp}}_{F}}={\left(\frac{\alpha_{p}(r)}{{{<{\delta n}_{i}(r)>}^{dp}}_{F}}\right)}^{\frac{{∆}_{p}-{∆}_{p+dp}}{{∆}_{p-dp}-{∆}_{p}}},\;\mathrm{using}:\;\gamma^{dp}=\frac{{∆}_{p+dp}-{∆}_{p}}{{∆}_{p}-{∆}_{p-dp}}$$

Therefore, we could obtain:$$\alpha_{p+dp}(r)={{<{\delta n}_{i}(r)>}_{F}}^{\left(1-\gamma \right)}{\alpha_{p}(r)}^{\gamma }\;/\alpha_{p\to \infty }={<{\delta n}_{i}(r)>}_{F}$$

When generalizing, we obtained: (10)$$\alpha_{p+dp}\left(r\right)={{<{\delta n}_{i}\left(r\right)>}_{F}}^{\left(1-\gamma^{dp}\right)}{\alpha_{p}\left(r\right)}^{\gamma^{dp}}$$
where $\alpha_{p\to \infty }={<{\delta n}_{i}(r)>}_{F}$.

In our study, we took dp=0.2 as a thresholding path.

## 5. Scaling Analysis

For the scaling analysis, we used the box-counting method, counting the grid cells affected by the studied structure.

This analysis allowed us to measure the fractal dimension $D_{f}$ ($D_{f}=\Delta_{1}$) of the 2D Cantor clustering of the points, for which we calculated the reversibility efficiency $\gamma_{geo}$ according to the geometrical method:$$\gamma_{geo}=\frac{D_{f}-\Delta_{\infty }}{d-\Delta_{\infty }}$$

In order to perform this calculation, first, the crest dimension had to also be calculated. We could then obtain the geometric efficiency of reversibility $\gamma_{geo}$.

### 5.1. Crest Dimension Measurement

First, we computed the crest dimension $\Delta_{\infty }$ for the simplest cases that corresponded to a population factor a=1. This meant that the number of points per block was constant across the iterations.

Since fluctuation and intermittency develop on the spatial support or “effective zone” between two extremities of the hierarchy of structures, we could deduce that in our 2D Cantor clustering, the crest presented with the densest and most clustered areas compared to the remainder.

Given that the clustering of points occurred across iterations in the form of blocks, exhibiting scale invariance, the most densely clustered zones in points corresponded to the last iteration. Moreover, since the areas among the dense blocks were neglected across the fluctuation levels due to the adopted thresholding in the entropic skin model, we could compute the crest dimension $\Delta_{\infty }$ (case *a* = 1) with the classical box-counting method at $r=l_{i}$ by counting the number of blocks at the last iteration i (the highest), as follows:(11)$$\Delta_{\infty }=-\frac{ln\left(N_{b,i}\right)}{ln\left(\frac{l_{i}}{l_{0}}\right)}=-\frac{ln{(4}^{i})}{ln\left(\theta^{i}\right)}=-\frac{\ln\left(4\right)}{\ln\left(\theta \right)}\;for\;a=1$$
where $N_{b,i}$ is the number of blocks at the last iteration i.

On the other hand, the phenomenon of the clustering of points in the blocks highlighted the crucial role of the topical interparticle scale. The scaling analysis fully reached the points generated until the last iteration i (the highest), such that the number of affected meshes $N_{F}$ was equal to the number of these points at small variable scales r relatively close to the topical scale $l_{i*}$.

From this perspective, we noticed that the scale analysis remained homogeneous up to the topic scale of the homogeneous distribution (iteration 0) $l_{*0}$; thus, the curve began to enter a transitory phase before taking on an asymptotic appearance, such that its slope at smaller scales tended towards $\Delta_{\infty }$ (see Figure 4).

To this end, when generalizing a≠1, we undertook the analytical study of the shape of the scaling analysis curve through the use of iterations among the topical scales $l_{*i}$ and ∀ i, particularly at small scales. We, therefore, introduced the local fractal dimension $\Delta^{\left(i\right)}$, corresponding to iteration i, among the topical scales $l_{*i-1}$ and $l_{*i}$.

Indeed, our adopted geometric attractor generated self-similar distributed blocks containing homogeneously distributed points. There were obviously crossovers among these blocks through the iterations, in which there was also a homogeneous distribution of points. During the counting of points, we had to distinguish between the two zones that were called the block zone and cross zone, as shown in Figure 5. The two areas formed the whole of the generated geometry.

Knowing that at the topic scale $r=l_{*i}$ the number of affected meshes was theoretically equal to the number of points accumulated until this iteration i. Therefore, we defined the equation of the total number of points $n^{\left(i\right)}$ thrown into the 2D Cantor clustering up to iteration i:(12)$$n^{\left(i\right)}={n_{c}}^{\left(i\right)}+{n_{b}}^{\left(i\right)}=n_{c,1}\sum_{j=0}^{i-1}{\left(4a\right)}^{j}+{{4^{i}a}^{i}n}^{\left(0\right)}a\neq 0$$
such that ${n_{c}}^{\left(i\right)}$ and ${n_{b}}^{\left(i\right)}$ represent, respectively, the number of points having accumulated at the cross zone and at the block zone at iteration i. Moreover, $n_{c,1}$ is the number of points that accumulated at the cross zone at the first iteration and $n^{\left(0\right)}$ is the total number of points at iteration 0, where a represents the populating factor (see Figure 3).

We understood that:$$\frac{n^{\left(i\right)}}{n^{\left(i-1\right)}}=\left(\frac{n_{c,1}\sum_{j=0}^{i-1}{\left(4a\right)}^{j}+{{4^{i}a}^{I}n}^{\left(0\right)}}{n_{c,1}\sum_{j=0}^{i-2}{\left(4a\right)}^{j}+{{4^{i-1}a}^{i-1}n}^{\left(0\right)}}\right)$$

For a very important number of iterations, we obtained (for more details see Appendix A):$$\underset{i\to \infty }{\lim}\frac{n^{\left(i\right)}}{n^{\left(i-1\right)}}==4a$$

Using $\frac{l_{i}}{l_{i-1}}=\theta$, $a=\frac{n_{b,i+1}}{n_{b,i}}$ and Equation (7) we obtained:$$\frac{l_{*i}}{l_{*i-1}}=\theta /\sqrt{a}$$

Therefore, we found:$$\underset{i\to \infty }{\lim}\Delta^{\left(i\right)}=-\underset{i\to \infty }{\lim}\left(\frac{\ln\left(\frac{n^{\left(i\right)}}{n^{\left(i-1\right)}}\right)}{\ln\left(\frac{l_{*i}}{l_{*i-1}}\right)}\right)=-\frac{ln(4a)}{ln(\theta /\sqrt{a})}$$

For a=1 and $\theta =\frac{1}{3}$, which was the case of a normal one-third 2D Cantor:$${\underset{i\to \infty }{lim}\Delta }^{\left(i\right)}=\frac{ln(4)}{ln(3)}=1.2618$$

This verified the shape of the curve, such that its slope at small scales tended towards the crest dimension $\Delta_{\infty }$, as seen in Figure 4.

We verified this result with the use of both analytical and graphical methods for different values of aϵ [1, 1.4] and θ ϵ [0.25; 0.43].

This motivated us to search for the topical scale from which the fractal analysis curve adopted an asymptotic behavior of slope $\Delta_{\infty }$ in order to define the range of scales allowing for an approximate measurement of the crest’s dimensions. Indeed, since the local fractal dimension completely left the $\Delta_{\infty }$ value from $l_{1*}$ up through the scales as shown in Figure 4, this highlighted the role of this scale. We, therefore, defined the crest scale range between the global scale $l_{0}$ and the topical scale at the first iteration $l_{1*}$. For this purpose, we called $l_{1*}$ the crest scale $l_{crest}$.

### 5.2. Bulk Dimension Measurement

Let us go back to the definition of the bulk, since it was the lowest extremity in the hierarchy of skins showing the least fluctuation, which included all the highest fluctuation levels. Therefore, the bulk extension presented the spatial support, where the whole studied distribution occurred. Furthermore, we noticed in our 2D Cantor clustering that the density of points increased through the iterations, indicating that the least-fluctuating structure corresponded to the first iteration. Therefore, we could say that from $r=l_{*1}$, as we moved down the scale, the analysis curve began to leave the bulk domain towards the crest domain (by descending with the scale) when the bulk was in the scale range between $l_{*1}$ and $l_{0}$.

To test our hypothesis, we measured the local fractal dimension at variable steps of scales $∆r(r)=r-l_{*i}$ by fixing the smallest scale end on the topical scale of the last iteration $l_{*i}$ and varying the other end of the scale r or by performing the measurement for each ∆r.

We noted that when ∆r was close to the value $l_{*1}-l_{*i}$; i.e., for $r=l_{*1}$; $ln(\frac{r}{4096})=ln(\frac{l_{*1}}{4096})$ (in the logarithmic representation), the local dimension began to touch values very close to $\Delta_{\infty }$ as it descended with the scale range ∆r and, hence, the scale r (see Figure 6). This verified our observation. Thus, the topical scale at the first iteration $l_{*1}$ exhibited a characteristic scale to determine the two scale ranges of the crest and body.

Therefore, to calculate the fractal dimension $D_{f}$ or (${∆}_{f}$) of the distribution, we measured the linear regression slope of the scaling curve over the range between what we called the crest scale $l_{crest}=l_{*1}$ and the integral scale $r_{0}=l_{0}$ as the distribution extended over the entire space (see Figure 7).

To sum up, the 2D Cantor clustering scale analysis was explicitly dependent on topical scales through successive iterations. It was, therefore, divided into three phases. First, it began with the homogeneous phase extending between $l_{0}$ and $l_{*0}$. In this phase, the analysis had a dimension equal to the Euclidean dimension (d = 2). Then, it continued in a transition phase between $l_{*0}$ and $l_{*1}$, where the local fractal dimension changed from the Euclidean dimension value to the one of the crest dimension ${∆}_{\infty }$. Finally, the scaling analysis entered the crest phase, occupying the scale range ${[l}_{*1},l_{*i}]$, for which the curve exhibited an oblique asymptote with a slope equal to the crest dimension ${∆}_{\infty }$ (see Figure 7).

The scale analysis in the homogenous distribution gave us the fractal dimension $D_{f}$, equal to the Euclidean dimension of plan d=2.

## 6. Statistical Analysis

### 6.1. Scale Exponent Measure

Since the structure functions exhibited a rate of fluctuation across scales and fluctuation levels following the scaling law ${<{\delta n}_{i}(r)}^{p}>\sim r^{z_{p}}$, and in order to perform the statistical analysis of the studied geometry, we measured the scaling exponents. For this purpose, we calculated the structure functions according to Equation (5), and represented them in logarithmic coordinates showing the scaling exponent $z_{p}$ as the slope. In the homogeneous case, we obtained straight lines with a low dispersion of points, having slopes equal to p, from which we found that $z_{p}$ varied as p ($z_{p}=p$) (see Figure 8).

In the case of the intermittent Cantor, a certain log-periodicity appeared in the moments $ln{<{\delta n}_{i}(r)}^{p}>$ (at the different scale $l_{*i}$) as a function of the normalized scale among the successive values of $l_{i}$ where $l_{i}=\theta^{i}l_{0}$(see Figure 9). Note that $z_{p}$ was different from p, unlike the homogeneous case. Some deviation from homogeneity occurred [50,56].

Note that to define the inertial range, we represented $\ln\frac{{<{\delta n}_{i}(r)}^{p}>}{{<{\delta n}_{i}(r)>}^{p}}$ as a function of $\ln\frac{r}{r_{0}}$, in the homogenous case; we observed that there was a scale range where $\ln\frac{{<{\delta n}_{i}®}^{p}>}{{<®(r)>}^{p}}$ was constant, which corresponded to the inertial range (see Appendix B).

### 6.2. Extended Self-Similarity in Geometric Clustering Model

In order to highlight the phenomenon of intermittence and its characteristics, we used the hypothesis of extended self-similarity previously seen in studies of intermittence in developed turbulence. This hypothesis assumed the existence of the extended self-similarity of structure functions through different levels of fluctuations [3,54]. Therefore, we assumed that any structure function of point numbers at fluctuation level p was a power law of any other structure function at fluctuation level q. Through representing $ln{<{\delta n}_{i}(r)}^{p}>$ as a function of ${ln<{\delta n}_{i}(r)}^{q}>$ and measuring a slope value $\frac{z_{p}}{z_{q}}$, we obtained the slope $\frac{z_{p}}{z_{q}}$ equal to $\frac{p}{q}$ in the case of homogeneity. A deviation in $\frac{p}{q}$ had to occur in the case of intermittency.

Indeed, we undertook the verification of this law for both the homogeneous and intermittent cases (see Figure 10).

### 6.3. Calculation of Statistical Reversibility Efficiency $\gamma_{stat}$

To compute the statistical reversibility efficiency, we used Equation (10). We presented $\alpha_{p+dp}(r)$ and $\alpha_{p}(r)$ in logarithmic coordinates for all scales on the same graph. Assuming that γ was constant for all the scales, we obtained a linear curve; its slope was $\gamma^{dp}$ (see Figure 11).

To ensure that we worked in a range of scales where intermittency developed, we avoided scales smaller than $l_{*I}$ as the last iteration topic scale; thus, the measure scale range of $\gamma_{stat}$ was [$l_{*i},l_{0}$]. In the homogeneous case, the statistical analysis gave a factor $\gamma_{stat}=0.997\approx 1$.

## 7. Results

### 7.1. Intermittency as Deviation to Homogeneity

As we mentioned at the beginning, our objective was to propose a simple fractal geometric model to explain the statistical intermittency phenomenon. We applied the entropic skin theory concept and the extended self-similarity hypothesis to verify the ability of the proposed model to describe this phenomenon.

Indeed, we accomplished the statistical and geometrical analyses by performing all the measurements required for different degrees of intermittency through varying the intermittency intensity chosen as the symmetry scale θ. It was the most effective factor that explicitly and proportionally presented the density of intermittency. The parameter θ varied within a range of [0; 0.5], such that if θ tended to 0, the 2D Cantor’s clustering became highly intermittent. In addition, if θ neared the value of 0.5, the clustering became less intermittent and affected the homogeneity in such a way that the fluctuation to the average strongly decreased.

We measured the values of $\frac{z_{p}}{z_{1}}$ as a function of p by applying the self-similarity assumption. We found that the curve showed a linear shape very close to p, identical to the homogeneous case, such that $\frac{z_{p}}{z_{1}}∼p$ signified the existence of self-similarity; the structure function ${<{\delta n}_{i}(r)}^{p}>$ was a power law of $<{\delta n}_{i}(r)>$. This was analogous to the Kolmogorov theory in developed turbulence. A certain deviation from the power law, beginning from *p* = 1.2, was observed in the cases of 2D Cantor clustering for different intermittency parameters θ (symmetry scale). This highlighted the intermittent characteristics of the clustering as a structure linking the levels of fluctuations (see Figure 12).

The deviation from homogeneity increased inversely, proportional to the intermittency parameter θ. Moreover, when θ increased the ratio $\frac{z_{p}}{z_{1}}$, self-similarity approached the homogeneous case. This was compatible in the case of intermittency in fully developed turbulence.

### 7.2. Crest and Bulk Dynamic

The measurement of the bulk and crest dimensions for various intermittence parameters θ revealed the correspondence and harmonization between θ and the two dimensions, explicitly highlighting the density of intermittence.

We represented $D_{f}$ as a function of $\frac{1}{ln\theta }$ for $\Delta_{\infty }$, according to Equation (11). Figure 13 shows the variation in $D_{f}$ as a function of θ, which seemed to be parabolic, so that $D_{f}$ could grow with θ. The same applied for $\Delta_{\infty }$, while $\Delta_{\infty }(\frac{1}{ln\theta })$ was linear with the slope equal to ln4.

The interpolation of the two-dimensions curves $D_{f}=f(\frac{1}{ln\theta })$ and $\Delta_{\infty }=f(\frac{1}{ln\theta })$ gave two equations:(13)$$D_{f}=-0.15{\left(\frac{1}{ln\theta }\right)}^{2}-0.44\frac{1}{ln\theta }+1.7$$
(14)$$\Delta_{\infty }=-\frac{ln4}{ln\theta }$$

The two curves intersected at an abscissa $\frac{1}{ln\theta }=-1.44$; hence, θ≈0.5 and $D_{f}=\Delta_{\infty }\approx$ 2. This in fact corresponded to homogeneity.

### 7.3. Equality of $\gamma_{stat}$ and $\gamma_{geo}$

The obtained results showed equality between the statistical reversibility factor $\gamma_{stat}$ and the other geometric $\gamma_{geo}$, with a maximum relative error margin of 1.7%.

The two intermittency coefficients $\gamma_{geo}$ and $\gamma_{stat}$_,_ as functions of factor θ, showed almost the same trends, with negligible errors for both curves. They increased with θ up to a value of 1 for θ = 0.5 and then decreased, showing the increase in the intermittency intensity. Thus, this equality remained validated with a narrow margin of error. Indeed, the plot of $\gamma_{geo}(\theta )$ and $\gamma_{stat}(\theta )$ showed a clear correlation between these two quantities (see Figure 14).

To verify this equality, we performed the experiment by varying all the parameters that could influence the distribution, such as the symmetry scale θ, the number of points thrown per block $n_{b}$, the box size $l_{0}$ and the population factor a. However, we presented the ratio $\frac{\gamma_{geo}}{\gamma_{stat}}$ as a function of intermittency intensity θ. The values were scattered around the right-hand side: y = 1 (see Figure 15).

## 8. Conclusions

In this paper, we proposed a simple fractal geometric model to explain the statistical intermittency phenomenon. For this aim, we were inspired by the E.S.G entropic skin theory, characterizing the phenomenon of intermittency in fully developed turbulence using coupling fractal geometric and statistical intermittency properties.

For this aim, a particular geometric model of a cluster of points approaching the Cantor shape in 2D, with the symmetry scale θ being an intermittency parameter, was proposed. This model involved the introduction of point density differentiation across scales in a multiscale mechanism that reproduced several features of the intermittency phenomenon. To verify its ability to describe intermittency, we applied this model to the concept of the entropic skin theory, obtaining a conceptual validation.

This validation was confirmed through the intrinsic equality of the two reversibility factors $\gamma_{geo}$ and $\gamma_{stat}$, calculated using two different analysis methods: geometric and statistical. This equality was accompanied well by a correlation as a function of the symmetric scale θ for different parameter values that could influence the geometry.

We found that the intermittency phenomenon in the 2D Cantor cluster was adequately described with the multiscale dynamics, proposed by the entropic skin theory and coupling the fluctuation levels that extended between two extremes: the bulk and the crest. The dimensions $\Delta_{\infty }$ and $D_{f}$, as well as γ, explicitly expressed the intensity of cluster intermittency.

The crest displayed the densest and most aggregated areas with respect to the remainder of clusters, while the bulk displayed the spatial extension where all the studied clusters took place.

Then, at the statistical analysis level, we applied the extended self-similarity hypothesis (E.S.S.) to the model; a deviation from the homogeneity assumed by Kolmogorov in the fully developed turbulence was noticed. This highlighted the intermittency dynamics as a multiscale structure linking the levels of fluctuations.

One of the potential applications of our work is related to wall turbulence, which needs better understanding of the involved phenomena. Our approach is related to experimental near-wall turbulence. The study of the phenomenon of intermittency in wall turbulence based on ESG and the notion of scale entropy used to validate the clustering fractal model was applied to an experimental database of boundary layer flows [3]. This geometrical approach was validated with experimental data [57] considering a set of 50 cases corresponding to 50 values of y+ ranging from 1 to 2722. These boundary layer experiments showed the interest in our approach for a wide range of wall distances (1 ≤ y+ ≤ 2722). Using extended self-similarity, the scaling exponents were found to vary, with y+ displaying a tendency toward saturation in the vicinity of the wall, a behavior captured with the ESG. The theoretical and experimental results showed that the entropic skin geometry based on conceptual yet simple geometrical arguments could describe intermittency features of wall turbulence.

The aim of our study is to find potential applications in computational fluid dynamics (CFDs). For RANS and hybrid RANS/LES modeling of wall-bound turbulent flows, the demand for a fine grid resolution is great near walls, particularly for high Reynolds number flows. Hybrid RANS/LES, used in the outer region large Eddy simulation (LES) approach and in the near-wall region low-Re RANS model, remains computationally intensive and time consuming due to the high demand for wall-normal grid resolutions. It is possible to decrease the demand for fine grid resolutions at the wall region by using wall functions. In some cases, these methods have shown that they allowed to obviate the need for solving the turbulent kinetic energy (TKE) equation near walls, resulting in quicker and more accurate engineering computations [58,59,60]. Our proposed fractal model could present an interesting framework for the development of new tools for modeling near wall turbulence, especially since it highlighted the fractal aspect of statistical intermittency.

This study also provides a basis for various potential future applications for complex systems and for ecological and social phenomenon.

## Figures and Tables

**Figure 1 entropy-25-00749-f001:**
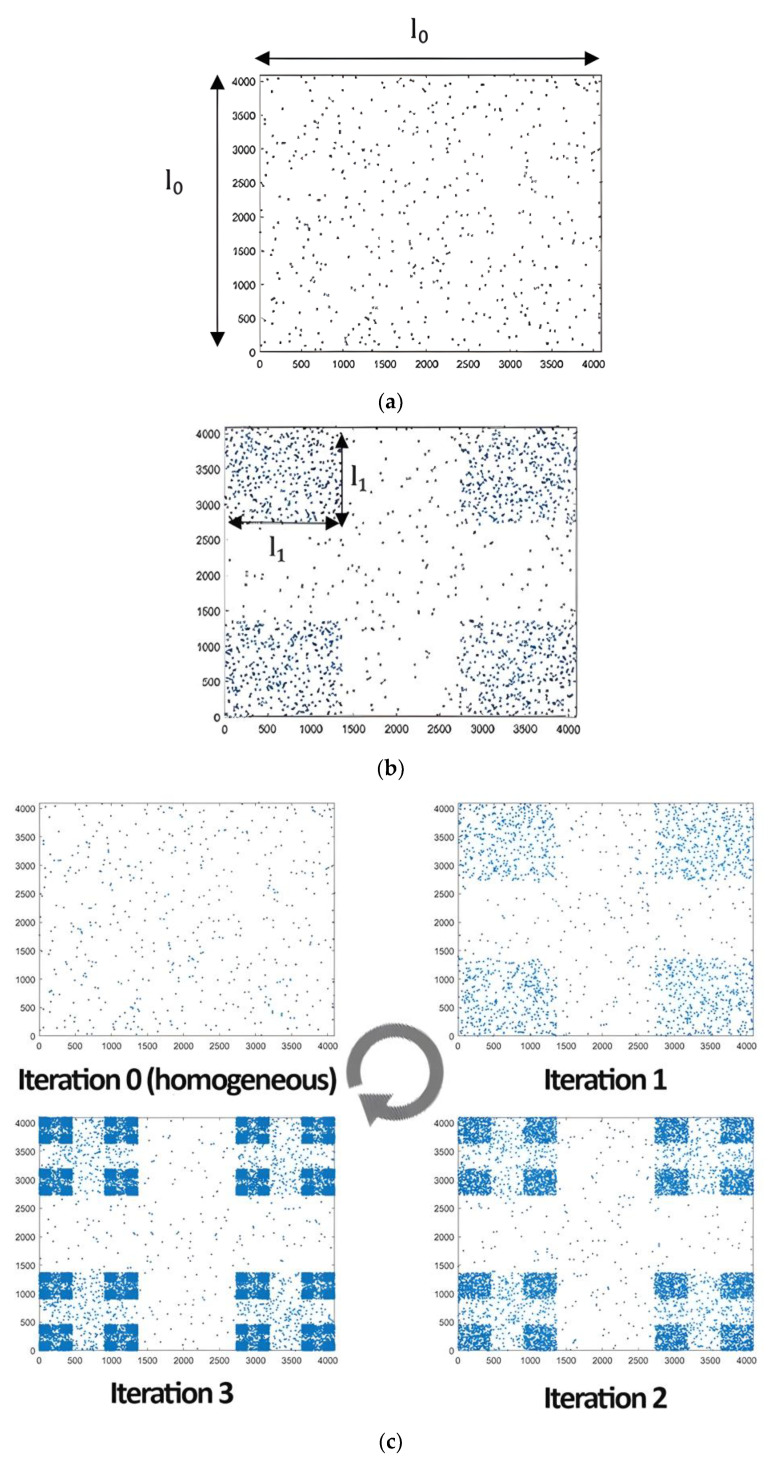
Two-dimensional Cantor clustering model structure: (**a**) step 1, (**b**) step 2 and (**c**) clustering model across iterations (θ = 1/3).

**Figure 2 entropy-25-00749-f002:**
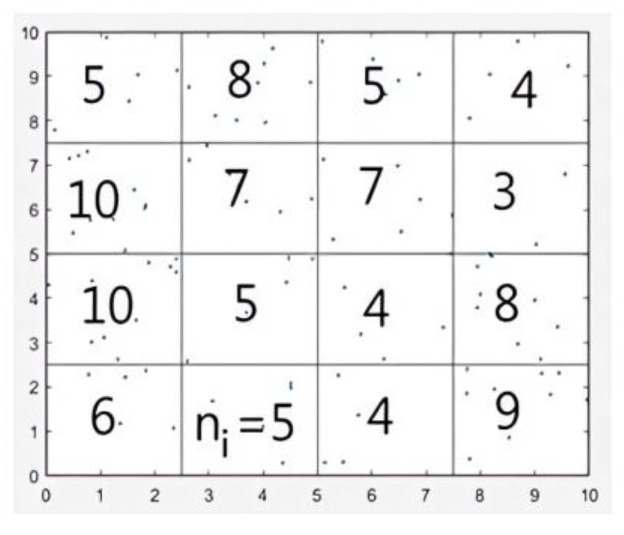
Covering the square box with a grid.

**Figure 3 entropy-25-00749-f003:**
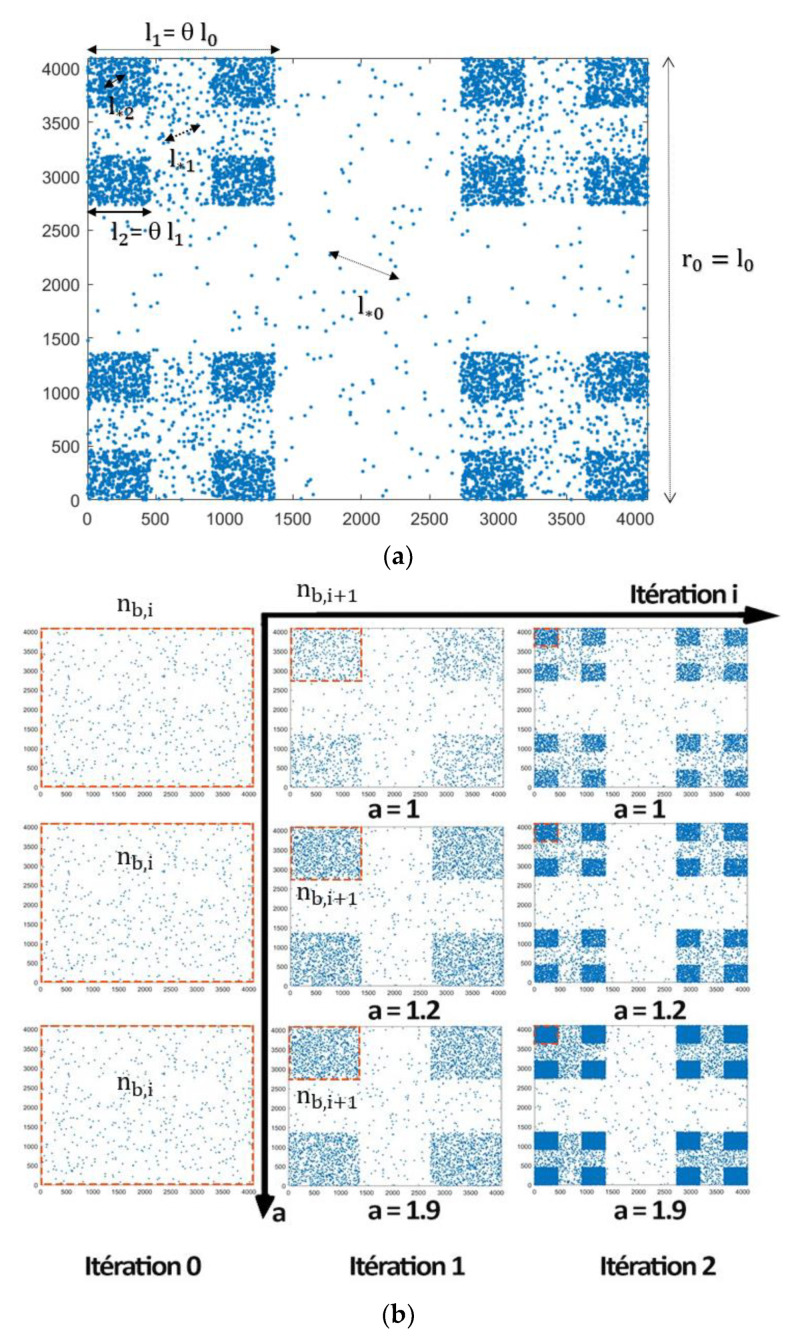
(**a**) The topic scale $l_{*i}$ and block size across iterations $l_{i}$; (**b**) the population factor $a=\frac{n_{b,i+1}}{n_{b,i}}$.

**Figure 4 entropy-25-00749-f004:**
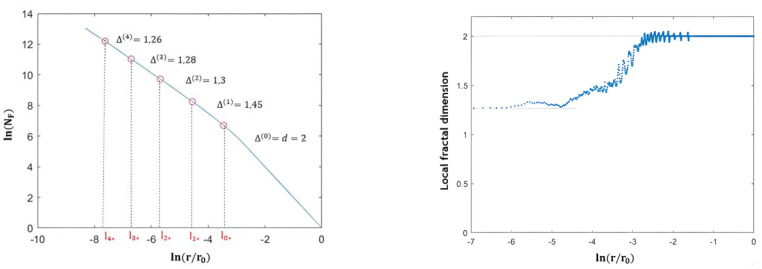
On the left: scaling analysis of a 2D Cantor clustering of one-third (θ = 1/3) with population factor a = 1 in a box of 4096 × 4096-sized pixels, reaching 4 iterations: the fractal dimension between topical scales across iterations $\Delta^{\left(i\right)}$ started to take on the value of $\Delta_{\infty }$ since the topical scale $l_{3*}$ of the third (the second to last) iteration. On the right: there was variation in local fractal dimension as function as scale r (step of scale equal to 6) as it increased with scale from 1.26 to 2; the local fractal dimension started to completely leave the value of $\Delta_{\infty }$ = 1.26 from scale $l_{1*}$.

**Figure 5 entropy-25-00749-f005:**
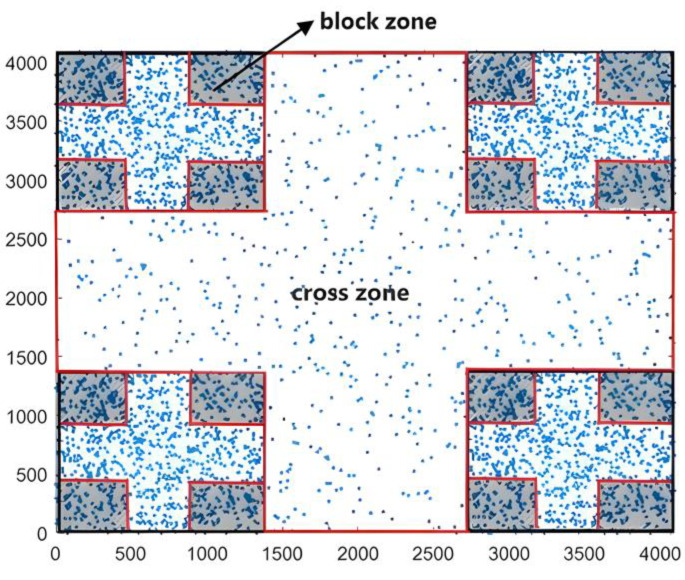
Cross zones (framed zones in red) and block zones (hatched zones) in the 2D Cantor clustering model with a surface area of 4096 × 4096 pixels. There were 16 block and 5 crosses.

**Figure 6 entropy-25-00749-f006:**
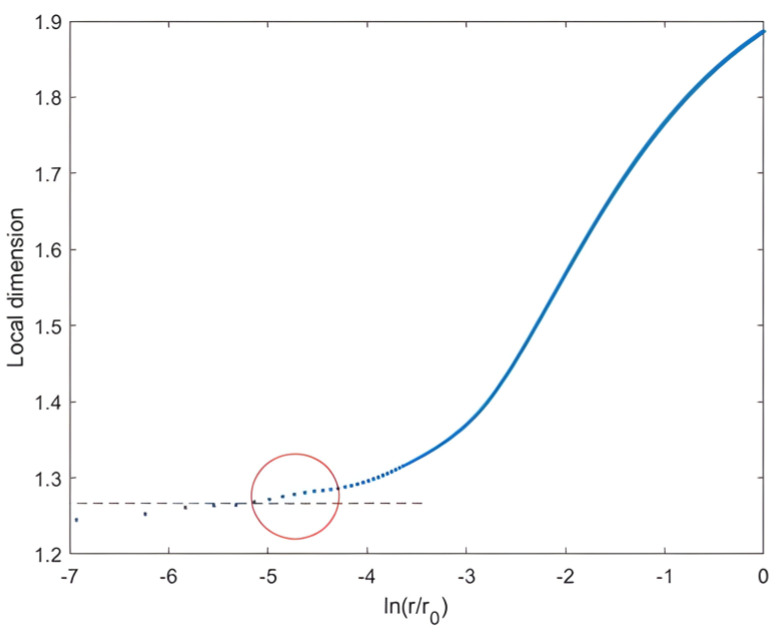
The local fractal dimension as a function of scale r (logarithm) at the variable step $∆r={r-l}_{*i}$ for 2D Cantor distribution one-third $\left(\theta =\frac{1}{3}\;and\;a=1\right)$. At the scale range $ln(\frac{∆r}{4096})$ near to $ln\left(\frac{{l_{*1}-l}_{*i}}{4096}\right)$, i.e., ($r=l_{crest}=l_{*1}\approx 43$ ), the local dimension started to reach values very close to ${∆}_{\infty }$ = 1.26.

**Figure 7 entropy-25-00749-f007:**
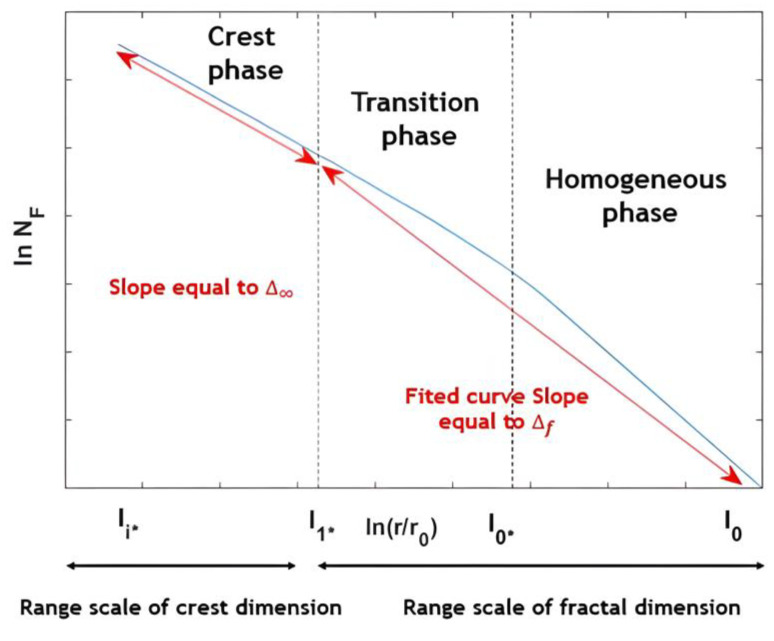
The three phases of the 2D Cantor distribution scale analysis: the bulk dimension represents the linear slope of the curve between the global scale $r_{0}$ and the crest scale ${l_{crest}=l}_{*1}$.

**Figure 8 entropy-25-00749-f008:**
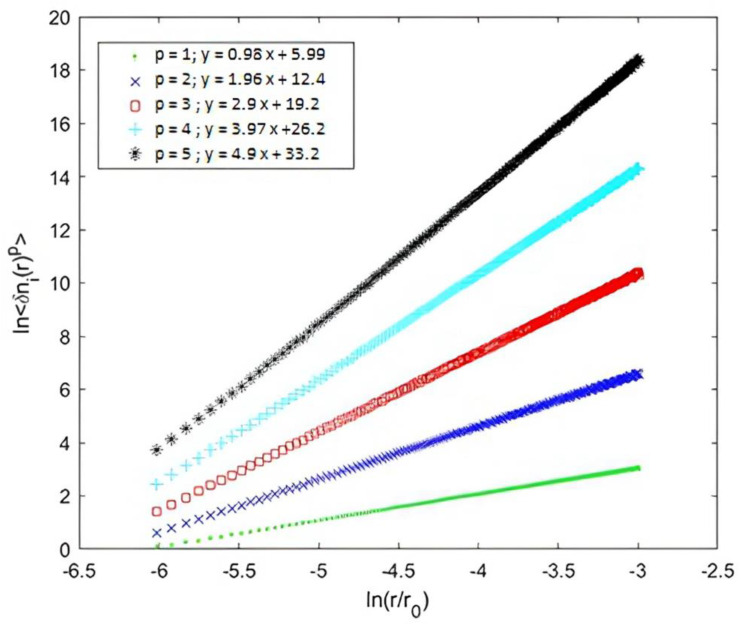
Logarithmic representation of structure functions ${<{\delta n}_{i}(r)}^{p}>$ of orders 2, 3, 4 and 5 as functions as the scale r at scale range for (ln(10/4096)<ln(r/4096)<ln(205/4096)), where ${<{\delta n}_{i}(r)}^{p}>∼\sim r^{z_{p}}$. We found that $z_{p}=p$ in the homogeneous case.

**Figure 9 entropy-25-00749-f009:**
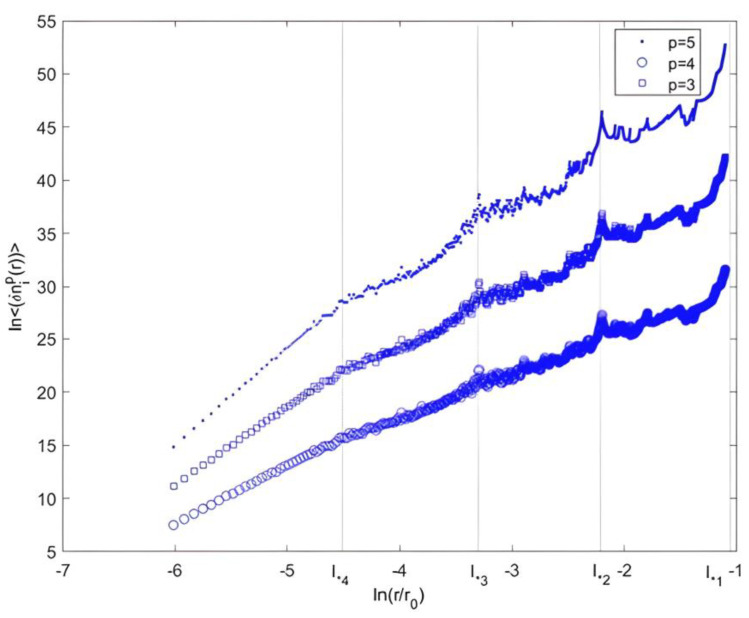
Structure function ${<{\delta n}_{i}(r)}^{p}>$ of different orders 3,4 and 5 for a 2D Cantor clustering $\left(\theta =\frac{1}{3}\right)$, vs. the normalized scale.

**Figure 10 entropy-25-00749-f010:**
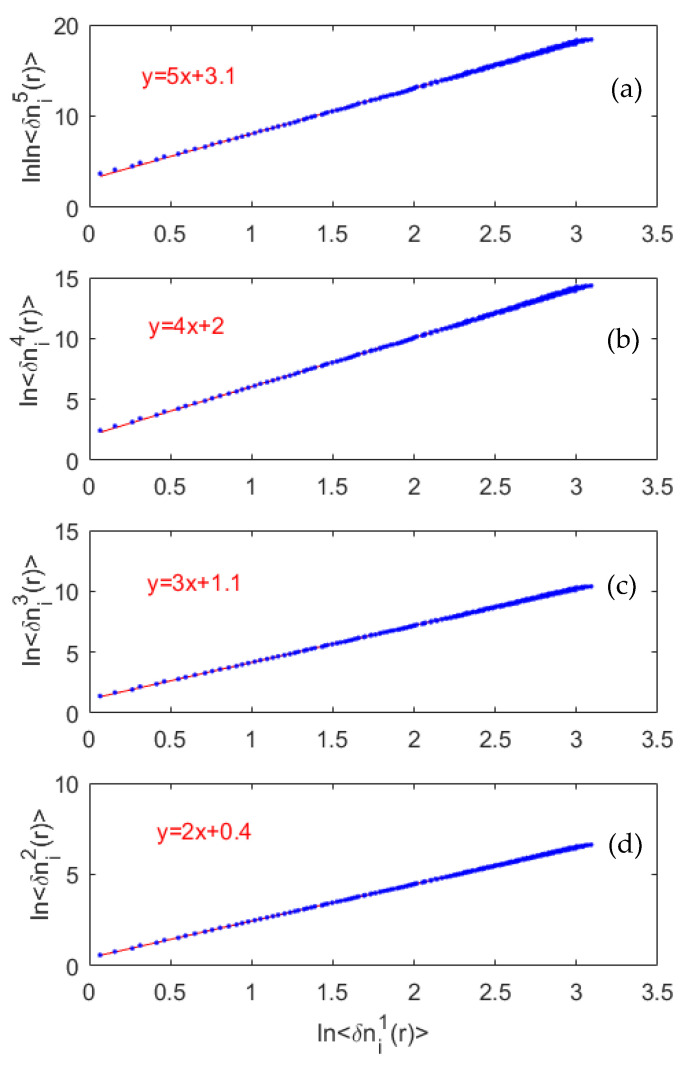
Logarithmic representation of structure functions of orders 5 (**a**), 4 (**b**), 3 (**c**) and 2 (**d**) as a function of a structure function of order 1 at scale range for (ln(10/4096)<ln(r/4096)<ln(205/4096)) in case of homogeneity.

**Figure 11 entropy-25-00749-f011:**
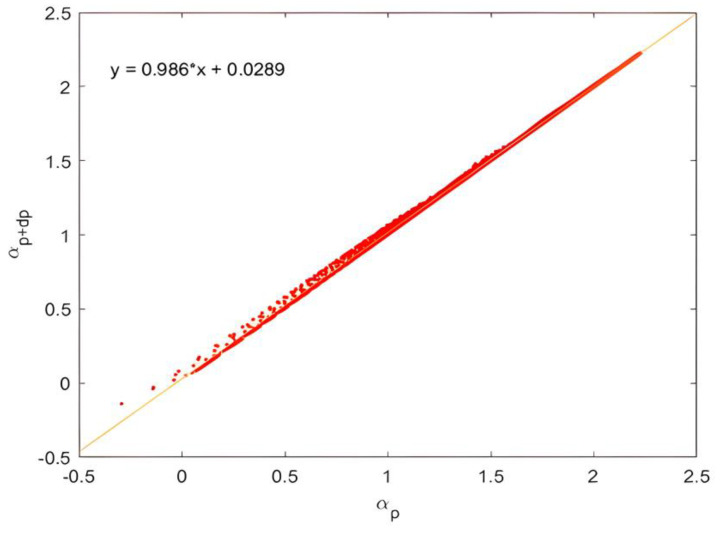
Logarithmic representation of $\alpha_{p+dp}(r)$ as a function of $\alpha_{p}(r)$. The gamma measurement scale range was [$l_{*i}$, $l_{0}$]. The obtained slope represented $\gamma^{dp}$.

**Figure 12 entropy-25-00749-f012:**
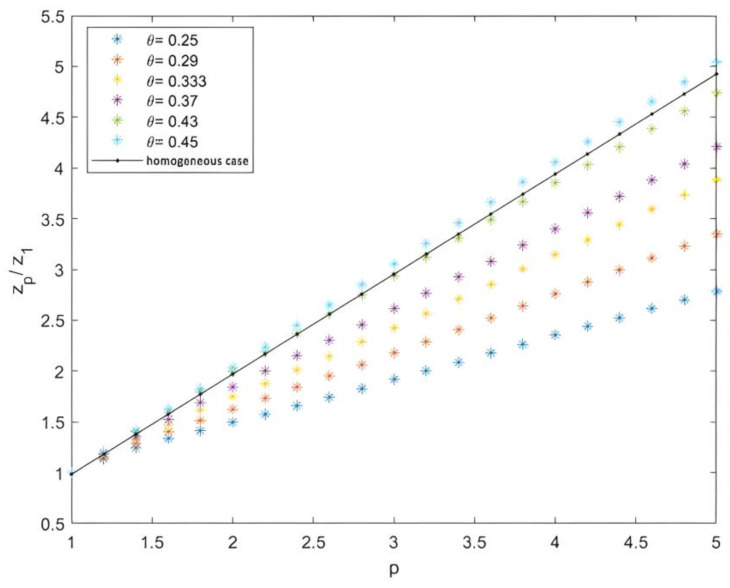
Scaling exponent ratios as a function of fluctuation levels p.

**Figure 13 entropy-25-00749-f013:**
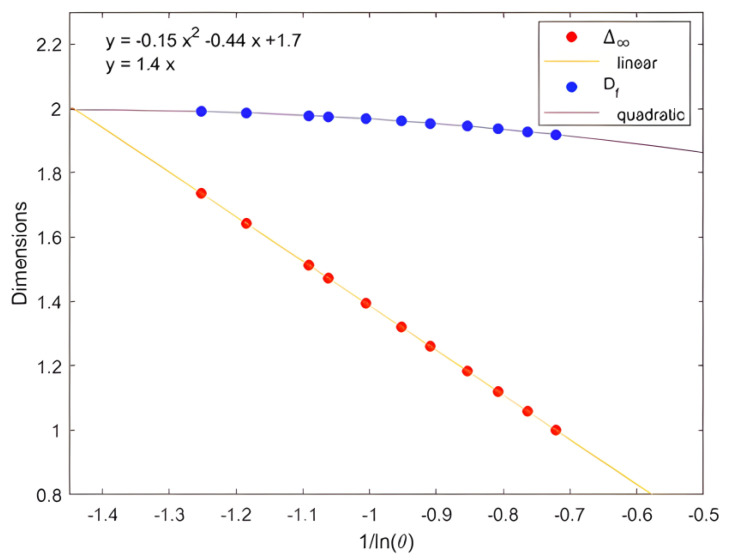
Bulk dimension $D_{f}$ and crest dimension ${∆}_{\infty }$ as a function of $\frac{1}{ln\theta }$. Both curves converged at Cantor scale value θ≈0.5, where $D_{f}={∆}_{\infty }\approx d$.

**Figure 14 entropy-25-00749-f014:**
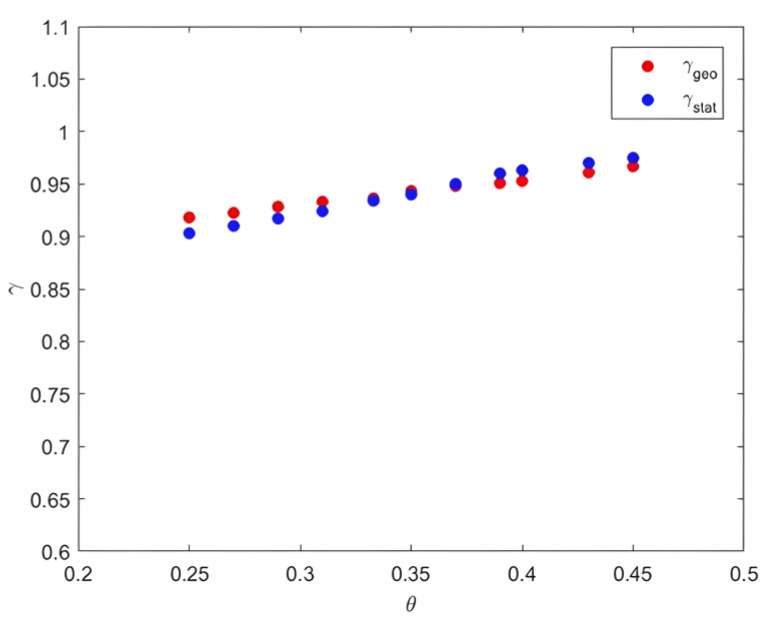
$\gamma_{geo}$ and $\gamma_{stat}$ as functions of Cantor scale θ.

**Figure 15 entropy-25-00749-f015:**
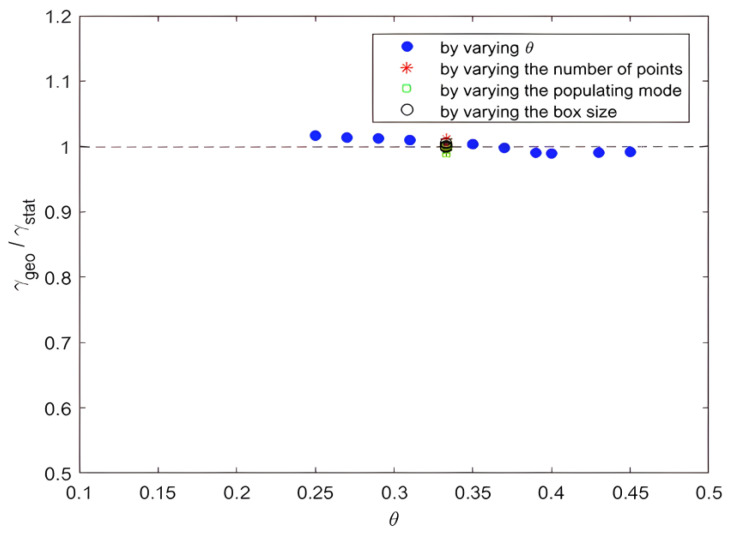
$\frac{\gamma_{geo}}{\gamma_{stat}}$ reported as a function of intermittence intensity θ.

## Data Availability

Not applicable.

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
