# Peer review of "Fractal Geometric Model for Statistical Intermittency Phenomenon"

_entropy, 2023, doi:10.3390/e25050749_

Round 1

Reviewer 1 Report

The presented work, like all previous articles of the authors on this topic, does not have the necessary motivation in the analysis of Kolmogorov's works. Here, the authors need to study at least such literature as

A. S. Monin and A. M. Yaglom, Statistical Fluid Mechanics, M.I.T. Press (1971). This book was written by Kolmogorov's students.

In addition, based on the materials, the conclusion suggests itself that the authors are not experts in the field of turbulence, since no one has ever perceived the 5/3 or 2/3 laws as universal, in the part that, depending on the Reynolds value, this inertial region has a different wavenumber range. And most importantly, the authors did not pay any attention to the area of turbulence generation. Therefore, introduction 1.1-1.2 needs to be redone thoroughly.

Further, the authors introduce quantities that do not try to connect with the entropy or Kolmlgorov-Sinay or Shannon or Rennie, which are the basis for the construction of a dynamical system. In this regard, the author needs to show the fundamental difference between the quantities introduced by them with the apparatus and the correlation dimension of attractors, according to the well-known works of the theory of strange:

50. Malraison, B., Berge, P., Dubois, M.: Dimension of strange attractors: an experimental determination for the chaotic regime of two convective systems. J. Phys. Lett. 44, L897–L902 (1983)

51. Procaccia, I., Grassberger, P.: Characterization of strange attractors. Phys. Rev. Lett. 50, 346–349 (1983)

52. Procaccia, I., Grassberger, P.: Estimation of the Kolmogorov entropy from a chaotic signal. Phys. Rev. A 28(4), 2591–2593 (1983)

53. Procaccia, I., Grassberger, P., Hentschel, H.G.E.: Dynamical Systems and Chaos, pp. 212–222. Springer, Berlin (1983)

 54. Grassberger, P., Procaccia, I.: Measuring the strangeness of strange attractors. Phys. D Nonlinear Phenom. 9(1), 189–208 (1983)

55. Grassberger, P., Procaccia, I.: Dimensions and entropies of strange attractors from a fluctuating dynamics approach. Phys. D Nonlinear Phenom. 13(1–2), 34–54 (1984). https://doi.org/10.1016/0167-2789(84)90269-0

56. Gromov, P.R., Zobin, F.B., Rabinovich, M.I., Reiman, A.M., Sushchik, M.M.: Finite dimensional attractors in shear flow with feedback. Dokl. Akad. Nauk 292(2), 284 (1987)

57. Rabinovich, M.I., Reiman, A.M., Sushchik, M.M., et al.: Correlation dimension of the flow and spatial development of dynamic chaos in the boundary layer. JETP Lett. 13(16), 987 (1987)

In this regard, it is necessary to fundamentally change and supplement Section 2.

  In addition, in sections 2 and 3 there are no relations showing the difference between the results and the known work on intermittency performed on the theory of strange attractors.

  Obtained by the authors, some correspondence with the parameter introduced by them and the experimental distribution of intermittency does not have a physical justification and value, since it is given for an isothermal classical flow, which are now well studied for practice, and more complex flows have other distributions.

The work is in the nature of an exclusively mathematical selection of a function for the intermittency coefficient. At the same time, the article in the literature section lacks the most important sources in the field of turbulence, the theory of entropy and strange attractors. This is probably why inaccurate interpretations of the laws of the inertial region of the turbulence spectrum are given and the advantages of the introduced concepts of entropies are not shown in comparison with the entropies of Kolmlgorov-Sinay or Shannon or Rennie, which raises numerous questions.

Even for applying the results obtained in the field of RANS, RANS-LES , LES there is no justification, since in these numerical codes the wall functions are selected by varying numerous parameters.

The work needs to be fundamentally rewritten.

Author Response

Dear Reviewer 1

We were pleased to know, by a letter from the Editor of Entropy MDPI, that our manuscript was rated as potentially acceptable for publication in Entropy MDPI, subject to adequate revision and response to your comments.

Please, find the revised manuscript with corrections based on your comments.

Thank you very much for your valuable comments. We would like to thank you warmly for your efforts in reviewing our manuscript, especially for your interesting comments that helped us to improve our work.

We agree with your comments and we did our best to improve the manuscript in a short time. Taking into account the short time in responding and submitting the revised version of the manuscript, we were not able to rewrite the overall manuscript. We know that there are different theories based on fractal geometry (strange attractors, multifractal…) and our work represents one approach.

Response 1:

Kolmogorov’s theory of fully developed turbulence assumes that energy dissipation occurs through an homogenous and isotropic way (Kolmogorov 1941). Following Kolmogorov’s formalism, turbulent flows are usually studied using structure functions. Kolmogorov’s classical theory, which postulates an homogeneous energy dissipation (that should be space-filling), predicts universal scaling laws for high Reynolds numbers and scales belonging to the inertial range. Kolmogorov predicted that the exponents should follow a linear law. The case p = 2 corresponds to the Kolmogorov's spectrum.

Kolmogorov, A.N. The local structure of turbulence in incompressible viscous fluid for very large Reynolds numbers. Doklady Akademii Nauk SSSR 1941, 30, 9–13.

Kolmogorov, A.N. On degeneration (decay) of isotropic turbulence in an incompressible flow. Doklady Akademii Nauk SSSR 1941, 32, 538–540.

We all agree with you that the Kolmogorov theory only applies to homogeneous and isotropic turbulence. However, when we look at turbulence near the wall, we find that the pure Kolmogorov spectrum is not respected, showing deviations from the theory. Indeed, the turbulence near the wall shows deviations from the pure Kolmogorov spectrum. The presence of intermittency, i.e. deviation from the linear behavior, indicates that this is not pure Kolmogorov behavior. It is therefore important to take these deviations into account when studying turbulence, especially near walls, to understand the behavior of the associated complex phenomenon.

Response 2:

The link between entropic skin intermittency theory and the concept of multifractals was established in Queiros-Conde (2001). This multifractal theory is related to research on strange attractors in turbulence, as studied by Procaccia and Grassberger (1983)

Queiros-Conde, D. Internal symmetry in the multifractal spectrum of fully developed Turbulence. Phys. Rev. E 2001, 64, 015301(R).

Procaccia, I., Grassberger, P.: Characterization of strange attractors. Phys. Rev. Lett. 50, 346-349 (1983)."

Strange attractors were introduced to describe turbulence by simplified coherent structures. The spirit of the theory of strange attractors has allowed the recent development of the proper orthogonal decomposition (POD). Our work is a kind of a decomposition between two main sets (bulk and crest) in the hope to describe the turbulence by a simplified coherent structure. Our paper does not deal with the pure entropic skin theory which is only statistical by postulating the existence of fractal dimensions, but we propose a direct geometrical application of this statistical theory.

Response 3:

One of the potential applications of our work is related to wall turbulence which need better understanding of the involved phenomena. Our approach is related to experimental near-wall turbulence. The study of the phenomenon of intermittency in wall turbulence based on ESG and the notion of scale-entropy, used to validate the clustering fractal model, was applied to an experimental database of boundary layer flows [3]. This geometrical approach was validated with experimental data [57]: A set of 50 cases corresponding to 50 values of y+ ranging from 1–2722 has been considered. These boundary layer experiments have shown the interest of our approach for a wide range of wall distances ( 1≤ y+≤ 2722). Using extended self-similarity, the scaling exponents was found to vary with y+ and display a tendency to the saturation in the vicinity of the wall, a behavior which is captured by ESG. The theoretical and experimental results show that entropic-skins geometry based on conceptual but simple geometrical arguments can describe intermittency features of wall turbulence.

One important advantage of entropic-skins geometry is to establish a direct link between multi-scale features and statistics usually defined in this field such as structure functions and scaling exponents. Our geometrical framework is quite general and could be applied to a wide variety of systems involving multi-scale and complex scaling features.

Queiros-Conde, D.; Carlier, J.; Grosu, L.; Stanislas, M. Entropic-Skins Geometry to Describe Wall Turbulence Intermittency. Entropy 2015, 17, 2198-2217. https://doi.org/10.3390/e17042198

Carlier, J.; Stanislas, M. Experimental study of eddy structures in a turbulent boundary layer using particle image velocimetry. J. Fluid Mech. 2005, 535, 143–188.

The aim of our study is to find potential applications in computational fluid dynamics for wall-bounded turbulent flows where the demand of fine grid resolution is great, in particular for high Reynolds number flows, in the vicinity of the walls. The hybrid RANS/LES models allows to decrease the demand for fine grid resolution at the wall region. This can be achieved by using in the outer region Large Eddy Simulation (LES) approach and in the near-wall region wall functions or low-Re RANS model. These methods show the interest that they allow obviating the need for solving the TKE equation near the wall in the RANS, hybrid RANS/LES closures thus resulting in quicker and accurate engineering computations (Sundaravadivelu and Absi 2021). Our proposed fractal model could present an interesting framework for development new tools for modeling near wall turbulence. Especially since it highlights the fractal aspect of statistical intermittency.

Sundaravadivelu, K., Absi, R. Turbulent kinetic energy estimate in the near wall region of smooth turbulent channel flows. Meccanica 56, 2533–2545 (2021).

  1. Davidson, S. Dahlstrom, Hybrid LES-RANS: an approach to make LES applicable at high Reynolds number, Int. J. Comput. Fluid Dyn., 19 (2005), pp. 415-427
  2. Tucker, L. Davidson, Zonal k–l based large eddy simulation, Comput. Fluids, 33 (2004), pp. 267-287
  3. Temmerman, M. Hadziabdic, M.A. Leschziner, K. Hanjalic, A hybrid two-layer URANS-LES approach for large eddy simulation at high Reynolds numbers, Int. J. Heat Fluid Flow, 26 (2005), pp. 173-190.

As you notice, your comments were very useful and identified areas of manuscript that needed clarification. We would like to take this opportunity to express you our sincere thanks. We hope that this revised manuscript will be accepted for publication. We trust that it will bring new insights about this topic for the readers/community and will impulse constructive debate. We hope that this work, which is a part of PhD thesis of the first author, will be a first an important step for future research.

Sincerely Yours,

The authors

Reviewer 2 Report

The manuscript is devoted to the phenomenon of intermittency as a dynamic connection of fluctuation levels in a hierarchical context. This can be used for various applications for complex systems and social phenomena. A geometric model of points clustering approaching the Cantor shape in 2D, with a symmetry scale being an intermittency parameter, is proposed. To test the model's ability to describe intermittency, the authors applied the concept of Entropic Skins theory. Thus, a conceptual confirmation was obtained.

There are comments on the manuscript.

1. It is advisable to add a subsection on clustering and fractal analysis to the Introduction.

2. Section 3 considers a well-known approach. However, the presentation of the material required effort to understand it.

3. Should check the spelling of the formulas in the legend in Figure 8. 

Author Response

Dear Reviewer 2, 

We were pleased to know, by a letter from the Editor of Entropy MDPI, that our manuscript was rated as potentially acceptable for publication in Entropy MDPI, subject to adequate revision and response to your comments.

Please, find enclosed the revised manuscript with corrections based on your comments.

Indeed, We carefully considered your comments and responded in detail as follows: 

1- we added a subsection on clustering and fractal analysis to the introduction (in blue color).  

2- we added and improved some explanatory elements especially in figure 3.  

3- we checked the spelling of the formulas in the legend of figure 8 and did the necessary (in blue color).  

We would like to thank you warmly for your efforts in reviewing our manuscript, especially for your comments that allowed us to improve our work. 

We hope that this revised manuscript will be accepted for publication. We trust that it will bring new insights about this topic for the readers/community and will impulse constructive debate. We hope that this work, which is a part of PhD thesis of the first author, will be a first an important step for future research.

Yours sincerely 

The authors  

Reviewer 3 Report

Comments and Suggestions for Authors to the manuscript “Fractal geometric model for statistical intermittency phenomenon”:

1.       I suggest to shorten the Abstract and only mention the main outcomes of the research.

2.       In the Introduction I found quite good list of references, but I suggest to put additional references regarding intermittences in mechanical systems. I suggest to add 2 following papers:

Radial internal clearance analysis in ball bearings,
Maintenance and Reliability 23(1), pp. 42-54, 2021,

Detection of type of intermittency using characteristic patterns in recurrence plots, Physical Review E, 80, 026214, 2009

3.       In the end of Introduction please add the Remainder of the paper.

4.       All Figures have bad quality, please increase its DPI.

5.       Figure 7, how do you define various phases?

6.       Figure 10, I think that the style of axes description can be incorrect, it should be simplified.

7.       What are your future studies, please define the new directions.

I recommend the reviewed manuscript for publication after introducing above mentioned suggestions into the manuscript.

Author Response

Dear Reviewer 3, 

We were pleased to know, by a letter from the Editor of Entropy MDPI, that our manuscript was rated as potentially acceptable for publication in Entropy MDPI, subject to adequate revision and response to your comments.

Please, find enclosed the revised manuscript with corrections based on your comments.

Indeed, We carefully considered your comments and responded in detail as follows: 

1- We reduced the abstract by mentioning only the main results of the work. 

2- We added the 2 papers "Radial internal clearance analysis in ball bearings, Maintenance and Reliability 23(1), pp. 42-54, 2021" and "Detection of type of intermittency using characteristic patterns in recurrence plots, Physical Review E, 80, 026214, 2009", concerning intermittency in mechanical systems. They were very interesting, and they drew our attention to the recurrence plot (RP) which could be very interesting for our model. 

3- We added a paragraph at the end of the introduction concerning the remainder of the article (in red color). 

4- We improved the quality of the figures by increasing its DPI. 

5- About the definition of the different phases, we defined them as follows: 

The scale analysis is divided into three phases. First of all, it starts by the homogeneous phase which extends between and the topical scale of the homogeneous iteration. The analysis has in this phase a dimension equal to the Euclidean dimension (d=2).   

It then continues in a transition phase between and the topical scale of the first iteration, where the local fractal dimension changes from the value of the Euclidean dimension to that of the crest dimension. Finally, the scaling analysis enters the crest phase occupying the scaling range for which the curve exhibits an oblique asymptote with a slope equal to the crest dimension (see Figure 7).   

 We added explanatory material (paragraphs and sentences) to the ends of Sections 5.1 and 5.2 to clarify the context (in red color). 

6- We simplified the style of axes description in figure 10. 

7- We did add the perspectives and new directions of the work at the end of the conclusion section (in red color). 

We would like to thank you warmly for your efforts in reviewing our manuscript, especially for your comments that allowed us to improve our work. 

We hope that this revised manuscript will be accepted for publication. We trust that it will bring new insights about this topic for the readers/community and will impulse constructive debate. We hope that this work, which is a part of PhD thesis of the first author, will be a first an important step for future research.

Yours sincerely 

The authors  

Round 2

Reviewer 1 Report

The authors in the response claim that «Our proposed fractal model could present an interesting framework for development new tools for modeling near wall turbulence. Our proposed fractal model could present an interesting framework for development new tools for modeling near wall turbulence. Especially since it highlights the fractal aspect of statistical intermittency.»

The authors were invited to familiarize themselves with classical works in the field of turbulence. The article provides several graphs of comparisons. However, they have not made the main comparison in the boundary layer region, and this is strange, since it is well known that the intermittency coefficient near the wall is 1 see figure 1.

Fig.1 Distribution of the intermittency coefficient across the boundary layer

1.       These classic well-known experiments of 1955 are missing in articles .

2.       The article does not imply the possibility of matching your theory with Klebanov's results on the coefficient of intermittency in the boundary layer.

2.In addition, the paper does not provide formulas linking the entropy introduced by the authors with the entropy of Kolmlgorov-Sinai or Reni.

3. Also, the paper generally lacks at least theoretical estimates with the results of stochastic theory in turbulence.

See attached file.

Author Response

Dear Reviewer 

Thank you very much for your comments which allowed us to improve our manuscript. 

In order to improve the literature section, we added a well-known book in the field of turbulence: Monin, A.S.; Yaglom, A.M.; Lumley, J.L. Statistical Fluid Mechanics: Mechanics of Turbulence; The MIT Press.; 1979; Vol. 2; ISBN 978-0-262-13158-2. 

The theory of entropic skins, that has already been used to describe and to model the phenomenon of intermittency in turbulence, is a tool which aims to validate the model as the extended self-similarity (Queiros-conde 1999, 2000 ; Queiros-conde et al. 2015). Our manuscript does not deal with the pure entropic skin theory which is only statistical by postulating the existence of fractal dimensions, but we propose a direct geometrical application of this statistical theory. 

Comparison with Kolmlgorov-Sinay, Shannon or Rennie entropies is not the aim of the present manuscript. This could be a perspective of our future work. 

Finally, all the references have been carefully checked. 

As you notice, all the comments were very useful and identified areas of manuscript that needed clarification. We would like to take this opportunity to express our sincere thanks to you. We hope that the revised manuscript is accepted for publication in Entropy MDPI. 

Best regards, 

Reviewer 2 Report

The manuscript of the article has been revised. Some changes were made to the manuscript based on the comments of the reviewer. Explanations were given for the rest of the comments. All this allows us to conclude that the manuscript has changed for the better. Therefore, I consider it possible to recommend the manuscript for publication.

Author Response

Dear Reviewer 

Thank you very much 

Best regards, 

Reviewer 3 Report

After the revision of the manuscript I recommend the manuscript for its publishing in the present form.

Author Response

(The authors gave the same response as above.)
